# Deep Learning Denoising Improves and Homogenizes Patient [^18^F]FDG PET Image Quality in Digital PET/CT

**DOI:** 10.3390/diagnostics13091626

**Published:** 2023-05-04

**Authors:** Kathleen Weyts, Elske Quak, Idlir Licaj, Renaud Ciappuccini, Charline Lasnon, Aurélien Corroyer-Dulmont, Gauthier Foucras, Stéphane Bardet, Cyril Jaudet

**Affiliations:** 1Department of Nuclear Medicine, Baclesse Cancer Centre, 14076 Caen, France; 2Department of Biostatistics, Baclesse Cancer Centre, 14076 Caen, France; 3Department of Community Medicine, Faculty of Health Sciences, UiT The Arctic University of Norway, 9019 Tromsø, Norway; 4Department of Medical Physics, Baclesse Cancer Centre, 14076 Caen, France; 5ISTCT Unit, CNRS, UNICAEN, Normandy University, GIP CYCERON, 14074 Caen, France

**Keywords:** artificial intelligence, deep learning, CNN, [^18^F]FDG, PET, denoising, image quality, body habitus

## Abstract

Given the constant pressure to increase patient throughput while respecting radiation protection, global body PET image quality (IQ) is not satisfactory in all patients. We first studied the association between IQ and other variables, in particular body habitus, on a digital PET/CT. Second, to improve and homogenize IQ, we evaluated a deep learning PET denoising solution (Subtle PET^TM^) using convolutional neural networks. We analysed retrospectively in 113 patients visual IQ (by a 5-point Likert score in two readers) and semi-quantitative IQ (by the coefficient of variation in the liver, CV_liv_) as well as lesion detection and quantification in native and denoised PET. In native PET, visual and semi-quantitative IQ were lower in patients with larger body habitus (*p* < 0.0001 for both) and in men vs. women (*p* ≤ 0.03 for CV_liv_). After PET denoising, visual IQ scores increased and became more homogeneous between patients (4.8 ± 0.3 in denoised vs. 3.6 ± 0.6 in native PET; *p* < 0.0001). CV_liv_ were lower in denoised PET than in native PET, 6.9 ± 0.9% vs. 12.2 ± 1.6%; *p* < 0.0001. The slope calculated by linear regression of CV_liv_ according to weight was significantly lower in denoised than in native PET (*p* = 0.0002), demonstrating more uniform CV_liv_. Lesion concordance rate between both PET series was 369/371 (99.5%), with two lesions exclusively detected in native PET. SUV_max_ and SUV_peak_ of up to the five most intense native PET lesions per patient were lower in denoised PET (*p* < 0.001), with an average relative bias of −7.7% and −2.8%, respectively. DL-based PET denoising by Subtle PET^TM^ allowed [^18^F]FDG PET global image quality to be improved and homogenized, while maintaining satisfactory lesion detection and quantification. DL-based denoising may render body habitus adaptive PET protocols unnecessary, and pave the way for the improvement and homogenization of PET modalities.

## 1. Introduction

In recent years, the emergence of digital PET/CT with silicon photomultiplier (SiPM) detectors has been a leap forward in PET image quality (IQ) and lesion detectability, particularly for small lesions [1,2,3,4,5,6]. It has led to a decrease in injected activity and/or PET acquisition time. However, given the constantly increasing need for PET/CT examinations, patient throughput needs to be further optimized. During periods of high demand, we use an in-house PET protocol that has led to an 11% reduction in the time x activity product with a 33% reduction in the PET acquisition time compared to our standard PET protocol.

In this perpetual search to optimize PET procedures, image noise and degraded IQ may hamper interpretation, especially in patients with large body habitus. However, there is no consensus and very few reports on the relation between PET IQ and body habitus in the latest state-of-the-art PET with optimized time-of-flight (TOF) [7]. Yet efforts towards optimizing imaging in patients with larger body habitus are important in the light of the increasing prevalence of overweight and obesity worldwide [8].

Deep learning (DL), a subdivision of artificial intelligence (AI), has several applications in medical imaging including PET denoising [9,10,11,12,13]. Convolutional neural networks (CNN) are a suitable DL architecture [14]. Their use in other domains of medicine and in the environmental sciences is becoming widespread [15,16,17]. CNN may enhance PET image quality while reducing noise [18,19,20,21,22,23]. Two pilot studies have demonstrated the potential of CNN to improve and homogenize liver noise levels between patients [24,25].

Subtle PET^TM^ (Subtle Medical, Stanford, USA; provided in France by Incepto) is a CNN-based post-reconstruction PET denoising software for several radiopharmaceuticals including [^18^F]FDG, and has been approved by the Food and Drug Association (FDA) and accredited with European Conformity (CE) [26]. It uses a multi-slice 2.5D encoder-decoder U-Net as deep CNN architecture. Subtle PET^TM^ has shown promise for maintaining PET IQ when reducing significantly PET counts (ranging from −33% up to −75%), with very little impact on lesion detectability and quantitation [13,27,28,29]. Most [^18^F]FDG PET lesion radiomics features were found to be stable when applying Subtle PET^TM^ [30]. However, its use for improving IQ, in particular in patients with enlarged body habitus, and for homogenizing IQ between patients has received little attention until now. In a small group of 20 obese patients IQ scores were improved and became ‘good’ instead of ‘insufficient’ in 4/20 (20%) of patients by applying Subtle PET^TM^ on native PET without count reduction [27].

The present study had two aims: to investigate the relationship between native PET IQ and patient body habitus by a multivariable analysis; and to study the effect of Subtle PET^TM^ denoising on IQ. Subtle PET^TM^ was applied to native PET in the whole study population in an attempt to improve IQ and homogenize data between patients.

## 2. Materials and Methods

### 2.1. Patient Population

One hundred thirteen patients referred to our oncological institution for initial or follow-up [^18^F]FDG PET/CT between January and February 2020 were retrospectively included. Exclusion criteria were diabetes; major image motion artefacts; acquisition with arms beside the body; at least one hepatic metastasis in the past or present. The following patient body habitus data (i.e., weight, body mass index (BMI), and fat mass (FM)) were collected.

Body weight (kg) and height (m) were extracted from PET DICOM data and verified in the PET report. BMI was calculated as BMI (kg/m^2^) = weight/(height)^2^ [31] and fat mass (FM) as FM (kg) = weight (kg)—lean body mass (LBM, kg). LBM was estimated with the Janma formula, which is also suitable for very obese women [32,33]. Patient BMI categories were as follows: normal or underweight with BMI < 25 (*n* = 41; 36%); overweight with BMI ≥ 25 (*n* = 34; 30%); obesity with BMI ≥ 30 (*n* = 38; 34%), including 4 patients (4%) with severe obesity (BMI ≥ 40). These and other variables collected are shown in Table 1.

The study was approved by the institutional review board at the François Baclesse Comprehensive Cancer Centre and registered with the French Health Data Hub under reference I00160702202020. It was conducted in compliance with the French Research Standard MR-004 (Compliance commitment to MR-004 for the Centre François Baclesse No. 2,214,228 v.0, dated 3 July 2019). All patients received information and none expressed opposition to the use of their data.

### 2.2. Image Protocol

[^18^F]FDG PET/CT scans were performed in accordance with the EANM imaging guidelines [34]. Patients fasted for at least 6h before the intravenous [^18^F]FDG (4 MBq/kg) injection. Their weight was checked on a calibrated scale [35].

PET images from skull or skull base to mid-thigh were acquired between 55 and 65 min post-injection for 60 s per bed position on a digital PET/CT (VEREOS, Philips Healthcare, 2017). A 3D-ordered subset expectation maximization (OSEM) PET reconstruction was performed with time-of-flight (TOF) and Point Spread Function (PSF), using four iterations and four subsets, a 288 × 288 matrix and 2 × 2 × 2 mm^3^ voxel size. Scatter and attenuation corrections were applied. This ‘native PET’ protocol with a relatively low time × activity product of 4 (4 MBq/kg; 1 min/bed position) instead of 4.5 (3 MBq/kg; 1.5 min/bed position) in our regular clinical routine was implemented during periods when additional PET/CT exams needed to be performed for technical and/or organizational reasons.

Before each PET scan, a low-dose non-contrast-enhanced CT scan was acquired for attenuation correction and anatomical reference.

Denoising of native PET (‘denoised PET’) was performed by SubtlePET^TM^ software (version 1.0) on a local in-house server [26]. Blinded denoised images were returned automatically within two minutes to the viewing server (Syngo.via version VB30A, Siemens Healthineers, Erlangen, Germany).

### 2.3. Image Analysis

#### 2.3.1. Global Image Quality in Native and Denoised PET

Visual image quality

A 5-point visual global IQ score (Likert score) of all blinded native and denoised PET examinations, displayed side-by-side in Syngo.via, was attributed by two experienced readers. Scoring was defined as 5 = excellent, 4 = good, 3 = moderate, 2 = poor, 1 = very poor, where scores of 1 and 2 were considered unacceptable. This global score was based on PET liver heterogeneity, global image noise, normal structure contrast including the correct visualization of regions with no or little uptake (evaluated at intervertebral spaces).

Semi-quantitative image quality: analysis of the reference liver

A standard spherical volume of interest (VOI) with a 3cm radius was placed manually in the right liver lobe on native PET and automatically copied at the same position on denoised PET using ‘’3D Slicer’’ as software [36]. We avoided the upper hepatic region, main hepatic vessels, tissue boundaries and additional focal abnormalities (mostly cysts) on CT or PET. The VOI was adapted to a minimal 2.2 cm radius in 3% of patients with a small or narrow liver.

CV_liv_ as a semi-quantitative IQ parameter was measured in each liver VOI using the following formula:(1)CV (%)=100×Standard Deviation (SD)/SUVmean.

#### 2.3.2. Lesion Analysis in Native and Denoised PET

Visual lesion detectability

One reader classified each PET exam as pathologic or normal by the presence or not of [^18^F]FDG avid lesions. The number and ease of detection (related to visual contrast-to-background ratio) of all lesions with increased [^18^F]FDG uptake was compared side-by-side between native and denoised PET.

Semi-quantitative lesion analysis

In each patient, up to the five most intense [^18^F]FDG avid, malignant lesions in native PET were segmented semi-automatically, and the same lesions were segmented in denoised PET with PETTumorsSegmentation in 3D Slicer by an experienced nuclear medicine physician. SUV_max_ and SUV_peak_ in lesion VOI were extracted automatically with in-house software based on ITK [37].

### 2.4. Statistical Analysis

Variables are expressed as the mean +/− Standard Deviation (SD), and frequencies of visual IQ scores.

Visual IQ scores, CV_liv,_ liver SUV_mean_ and lesion SUV were compared between native and denoised PET by the Wilcoxon signed ranked test. Inter-reader agreement on IQ scores was evaluated by linearly weighted kappa [38]. Spearman’s rho (ρ) was used to analyse the correlation between visual IQ scores and other variables, in particular body habitus and CV_liv_. The association between CV_liv_ and separately weight, BMI and fat mass was tested by uni- and multivariable linear regression analysis in both native and denoised PET, comparing linear goodness of fit to exponential and quadratic transformations. Differences in Pearson correlation coefficients of CV_liv_ according to weight were analyzed using Fisher’s r to z transformation, and differences in slope coefficients by t-testing. For lesion detection, the rate of concordance between native and denoised PET was used. All tests were two-sided, with *p* < 0.05 considered to be statistically significant. STATA version 15 was used for analyses.

## 3. Results

### 3.1. Patient Population

In Table 1 we showed the characteristics of the 113 included patients.

**Table 1 diagnostics-13-01626-t001:** Patient and PET protocol data.

Gender	*n* (%)
female	77 (68%)
male	36 (32%)
Age (y) mean ± SD [range]	61.5 ± 13.5 (24–89)
Weight (kg)	74 ± 16 (35–110)
Height (m)	1.66 ± 0.10 (1.51–1.85)
BMI (kg/m^2^)	27 ± 6 (15–42)
Fat mass (kg)	26 ± 11 (5–55)
Glycemia (g/L)	1.01 ± 0.13 (0.70–1.38)
Injected ponderal activity (MBq/kg)	4.0 ± 0.2 (3.70–4.28)
Scan delay p.i. ^1^ (min)	58.3 ± 3.0 (55–65)
Bedposition scan duration (s)	60
PET indication *n* (%)	
Oncology (staging or follow-up)	95 (84%)
Breast	36 (32%)
Lung	17 (15%)
Other Gynecologic	14 (12%)
Other (lymphoma, anal, colorectal, bladder, thyroid, head and neck cancer, melanoma, myeloma or mixed)	28 (25%)
Characterization (benign vs. malignant): SPN ^2^	14 (12%)
Miscellaneous	4 (4%)

^1^ p.i.: post injection. ^2^ SPN: solitary pulmonary nodule.

### 3.2. Image Analysis

#### 3.2.1. Global Image Quality in Native PET

Visual image quality

In native PET, average IQ score by both readers was 3.6 ± 0.6, with scores of 2 (by at least one reader) in 4%, 3 in 43%, 4 in 46% and 5 in 7%. There was a significant negative correlation between IQ score and body habitus for both readers (*p* < 0.0001; Spearman ρ between −0.59 and −0.74 for each reader). This negative correlation was not significantly different for weight, BMI, and fat mass.

As shown in Figure 1A, in patients weighing at least 90 kg, 72.2% of native PET IQ scores attributed by at least one reader were moderate (3) and 11.1% poor or insufficient (2). IQ scores were better in lower weight categories. Categorizing by BMI, 76% of all obese patient IQ scores were moderate (3) and 22% poor or insufficient (2). There was no significant effect of gender, nor of other tested variables (age, study indication (initial vs. follow-up), glycemia, scan delay, and pathologic study) on visual IQ; *p* ≥ 0.8. Linearly weighted kappa showed fair agreement between both readers (κ = 0.35; 95% CI 0.22–0.45).

Semi-quantitative analysis

In native PET, CV_liv_ was 12.2 ± 1.6% and was associated with weight, BMI, and fat mass in univariable and multivariable linear regression analysis (*p* ≤ 0.0001). Figure 2 shows the univariable plot of CV_liv_ according to weight in native PET.

A significant association was found between male gender and CV_liv,_ in multivariable analysis including weight (*p* = 0.03), and it was more significant when testing together with BMI (*p* = 0.0003). Unassociated variables included age, study indication (initial vs. follow-up), glycemia, scan delay, and pathologic study. Visual IQ was negatively correlated with CV_liv_ (Spearman ρ = −0.63; *p* < 0.0001).

#### 3.2.2. Global Image Quality in Denoised PET: Improvement and Homogenization

Visual image quality

After PET denoising, visual IQ was significantly improved and was more homogeneous between patients (Figure 1B). IQ score was higher in denoised than in native PET, 4.8 ± 0.3 vs. 3.6 ± 0.6; *p* < 0.0001, with score 5 attributed in 80%, 4 in 19%, and 3 in 1%. Inter-reader kappa were not improved. There remained a significant negative correlation between visual IQ and body habitus (Spearman ρ = −0.42; *p* < 0.0001).

Semi-quantitative analysis

CV_liv_ was significantly lower in denoised than in native PET in all patients, 6.9 ± 0.9% vs. 12.2 ± 1.6%; *p* < 0.0001. After PET denoising, SD decreased considerably and liver SUV_mean_ increased moderately (liver SUV_mean_ of 2.8 ± 0.5 in denoised PET vs. 2.7 ± 0.4 in native PET; *p* < 0.0001). CV_liv_ was still significantly associated with weight (Figure 3), BMI and fat mass (*p* = 0.0001). Pearson’s correlation coefficients before and after denoising were not significantly different (*p* = 0.19 for CV_liv_ according to weight). However, the CV_liv_ slope according to weight was lower in denoised than in native PET (*p* = 0.0002). No additional association was found in multivariable analysis, in particular with gender (*p* = 0.16). 

Figure 4 illustrates the impact of body habitus on PET IQ and the improvement and homogenization of IQ after PET denoising in three patients.

#### 3.2.3. Lesion Analysis in Native and Denoised PET

Visual lesion detectability

In 47 patients, PET findings were normal with both PET modalities. In the remaining 66 patients, a total of 371 lesions with increased [^18^F]FDG uptake were detected. Of these lesions, 369/371 lesions (99.5%) were visualized with both modalities, while two small, low-activity bone lesions were detected only with native PET. In addition, ease of detecting the remaining concordant lesions was judged not significantly different in most of the lesions (359/369; 97.3%), better with native PET in 8/369 (2.2%) or better with denoised PET in 2/369 (0.5%). In Figure 5 we illustrate the effect of PET denoising on lesion detectability.

Semi-quantitative analysis of lesions

As shown in Figure 6, SUV_max_ and SUV_peak_ values in 101 analyzed lesions were significantly lower in denoised than in native PET (*p* < 0.001), with an average relative bias of −7.7 ± 5.6% for SUV_max_ and −2.8 ± 5.3% for SUV_peak._

## 4. Discussion

Visual and semi-quantitative native PET IQ values were lower in patients with larger body habitus on our digital PET/CT. Visual IQ was often moderate (43%) to even poor (3%), especially in patients with obesity and/or a high weight and fat mass, in whom a minority had a PET with good global IQ. A linear increase in the coefficient of variation in the liver according to body habitus was estimated with moderate curve fitting. In addition, there was a less significant association of CV_liv_ with male gender. Interestingly, CV_liv_ was strongly and negatively correlated with visual IQ.

Several reasons may explain the poorer PET IQ in patients with a large body habitus. Image noise is generated by increased soft-tissue attenuation (lowering count detection statistics), higher scatter, and possibly by an altered bio-distribution [39]. Moreover, obese patients may be more difficult to inject and to install on the camera table. This contributes to motion and other artefacts (mostly related to the arms) [40,41], although the latter occur less frequently with the more recent, optimized time-of-flight SiPM PET/CT cameras [42]. Only faint [^18^F]FDG uptake is visible in fat mass in a fasting state, and more in visceral than in subcutaneous fat [43,44]. Controversy continues about the influence of obesity on [^18^F]FDG uptake in fat tissue.

Higher liver noise in men than in women may be due to higher and more central abdominal fat levels. A higher incidence of steato-hepatitis in men [45], with possibly more heterogeneous and lower mean hepatic [^18^F]FDG uptake [46,47], is another putative explanation. Gender did not significantly influence visual scoring of global IQ in the present study.

The interpretive value and impact of higher noise levels are neither consensual nor predictable. Koopman et al. [5] found slightly higher average PET liver noise in digital than in analogue PET/CT (CV_liv_ = 14.7% vs. 13.3%) in the same patients, but also a higher ‘‘real lesion’’ detection. PSF modeling and a small voxel size (2 × 2 × 2 m^3^), as used in our PET reconstruction to optimize small lesion detectability, is known to affect image quality by increasing noise [4]. There is no consensus about how much noise in PET is acceptable or should be targeted. Nagaki et al. [48] and de Groot et al. [49] aim at liver noise levels of about 10%. Indeed, too much image noise can reduce exam accuracy (by increasing false positives and false negatives [13,50]), reading comfort and speed, and induce readers’ fatigue.

To increase efficiency and limit radiation, we decided to test a DL-based denoising solution in order to improve PET IQ and homogenize IQ between patients rather than implementing a PET protocol that is modified according to body habitus. Visual and semi-quantitative global [^18^F]FDG PET IQ improved significantly and became more uniform between patients by using Subtle PET ^TM^ in digital PET/CT. Visual IQ scores were excellent to good for nearly all patients in denoised PET, in contrast with native PET. Nevertheless, enlarging body habitus had still a negative effect on denoised PET IQ parameters.

Having less image noise did not lead to improved lesion detection or diagnostic accuracy in this study population, probably as native PET showed rather good lesion contrast-to-background ratios. Almost all lesions were detected with both PET modalities. On the other hand, the deep learning algorithm can encounter difficulties in distinguishing small lesions with low activity concentration from noise, leading to very few false negatives (0.5%) in denoised PET in this population. SUV_max_ and SUV_peak_ values of up to the five most intense lesions per patient were slightly lower in denoised PET, with a mean relative negative bias below 8% and very low (<3%) for SUV_peak_, as in previous studies [13,27]. By analyzing IQ, lesion detection and quantification after DL-denoising in this study population (with an 11% count reduced PET protocol) compared to 50% count reduced PET in our previous paper [13], it is possible to understand the effect of the DL-algorithm in less or more noisy conditions. Another finding in line with two previous studies is the slight increase in liver SUV_mean_ after PET denoising [13,27], which should be kept in mind while using the liver as a reference organ. In two other studies, no significant differences were found in native versus denoised liver SUV_mean_ [28,29]_._

This study has some limitations. First this was a retrospective analysis, so some errors in data sources or bias cannot be ruled out. Second, side-by-side visualization of both native and denoised PET series may have created a bias, but also allowed direct comparison of image quality [51] and visual lesion contrast. Another potential source of bias is the absence of lesions in many patients. This could decrease the generalizability of our findings. Moreover, we did not evaluate the readers’ ease of interpretation and visual comfort e.g., reading duration, subjective fatigue and overall diagnostic confidence. Finally, we are unable to compare our results to those of other more conventional denoising methods.

On the other hand, the study has some strengths. First, it lays the foundations for establishing the way in which PET image quality can be improved, optimized and homogenized. Second, a post-reconstruction DL-based PET denoising algorithm might be the best long-term practical solution to achieve this. Third, it would allow IQ to be homogenized between patients, and possibly also between different cameras and protocols. Finally, PET denoising by Subtle PET^TM^ might allow the acquisition time x activity product to be further decreased while maintaining IQ [13,27,28,29].

Striking the optimal balance between image quality, time x activity product and efficiency is of paramount importance for patients, clinicians and hospitals. Further improvement in DL algorithms would impact IQ favorably while preserving even better lesion detectability and quantification. This study is perhaps the first step in a move towards the homogenization and harmonization of PET by DL.

## 5. Conclusions

Given the constant pressure to increase patient throughput while respecting radiation dose, a native PET protocol with a relatively low time x activity product was implemented on our digital PET/CT.

We first studied visual and semi-quantitative [^18^F]FDG PET global image quality and their association with other variables, in particular body habitus, in native PET.

Native PET IQ was heterogeneous, often moderate and not satisfactory in all patients in this study population. Large body habitus and, to a lesser extent, male gender had a negative impact on PET IQ. To improve and homogenize IQ, deep learning based PET denoising by Subtle PET^TM^ was evaluated. Applied to native PET, DL-denoising improved global IQ substantially and satisfactorily while rendering it more uniform between patients. Lesion detection and quantification were preserved satisfactorily after PET denoising. DL-PET denoising may thus render body habitus adaptive protocols unnecessary. This study paves the way towards the improvement, homogenization and harmonization of PET by DL.

## Figures and Tables

**Figure 1 diagnostics-13-01626-f001:**
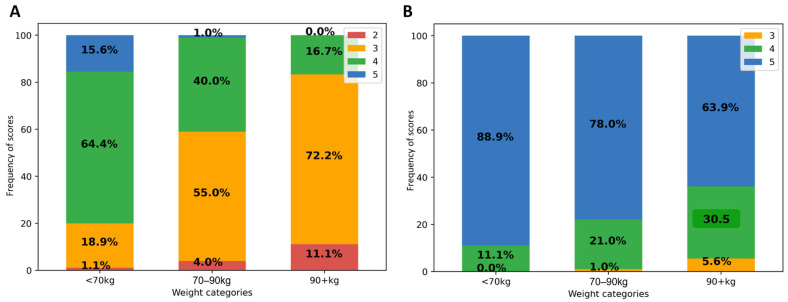
Impact of patient weight on visual IQ in native (**A**) and denoised PET (**B**). Legend: IQ score frequencies (in *Y*-axis) according to patient weight categories (on *X*-axis).

**Figure 2 diagnostics-13-01626-f002:**
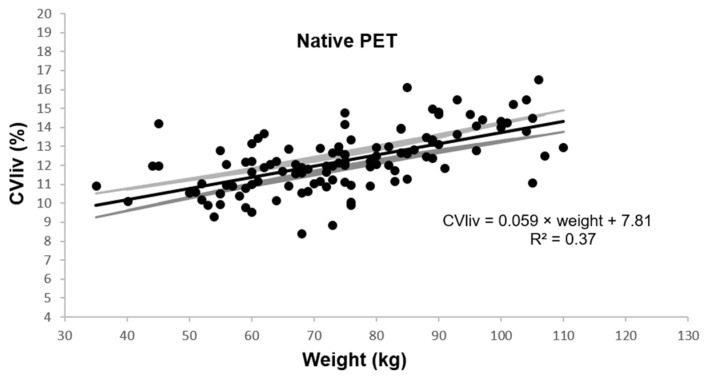
Coefficient of variation in liver according to patient weight. Legend: Estimated linear plot curve (continuous black line in the middle) with the 95% confidence interval (grey lines).

**Figure 3 diagnostics-13-01626-f003:**
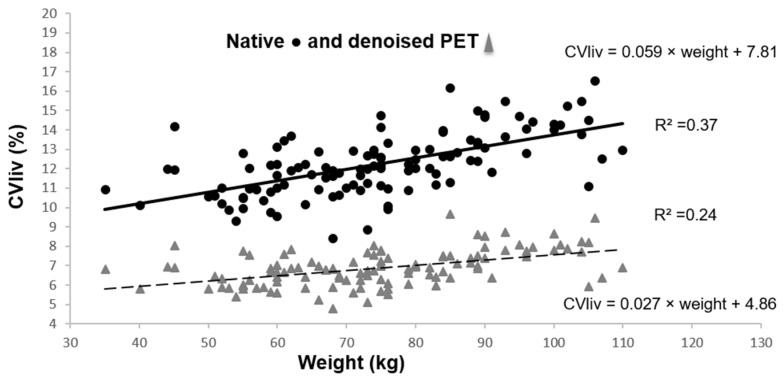
CV_liv_ according to weight in native and denoised PET. Legend: Continuous black line and dashed grey line show linear plot estimation.

**Figure 4 diagnostics-13-01626-f004:**
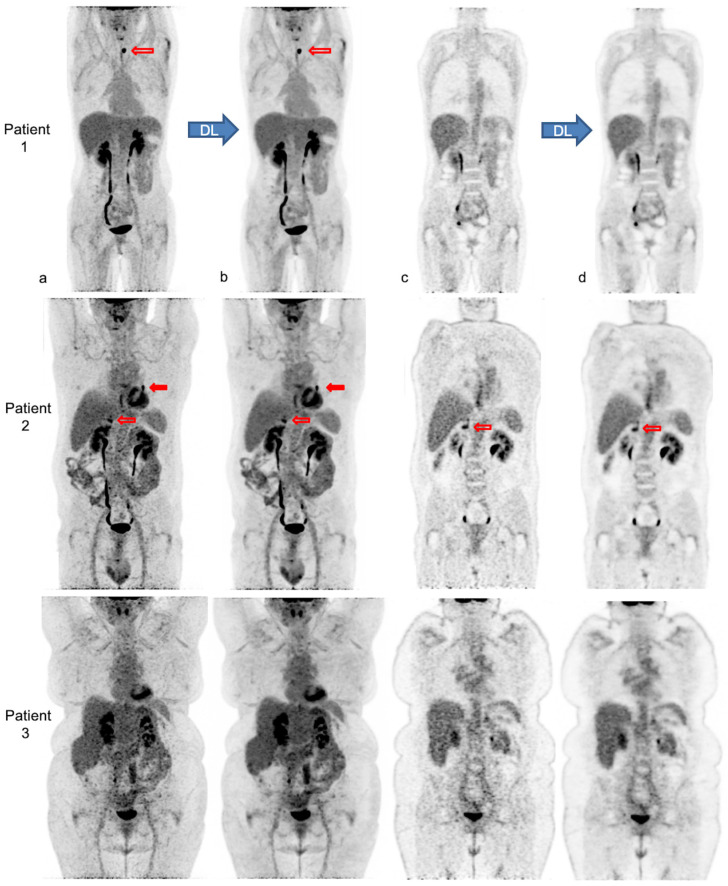
Native and denoised [^18^F]FDG PET images of three patients with different body habitus. Legend: MIP (**a**,**b**) and coronal views (**c**,**d**) of native PET (**a**,**c**) and denoised PET (**b**,**d**), respectively, are illustrated. DL: deep learning processing by Subtle PET^TM^. Patient 1 is a 36-year-old female with a weight of 55 kg, a BMI 22 kg/m^2^ and FM of 19 kg, scanned for a suspected paraneoplastic syndrome. The average visual IQ score with native PET was 4 versus 5 with denoised PET and CV_liv_ 9.9% vs. 5.8%, respectively. A [^18^F]FDG avid left thyroid nodule (red upper arrows), detected similarly in both PET series, proved to be a benign follicular adenoma. Patient 2 is a 63-year-old male with a weight of 89 kg, a BMI of 30 kg/m^2^ and FM of 27 kg, referred for lung cancer follow-up. IQ score with native PET was 3.5 vs. 4.5 with denoised PET and CV_liv_ 13.4% vs. 7.0%, respectively. The residual left lung lesion (full and upper red arrows) and a right adrenal metastasis (transparent and lower red arrows) were both detectable with native and denoised PET. Patient 3 is a 62-year-old female patient weighing 104 kg with a BMI of 38 kg/m^2^ and FM of 51 kg. She was scanned for cervical cancer follow-up. IQ score with native PET was 3 vs. 4 with denoised PET and CV_liv_ 15.5% vs. 8.2%, respectively. Both PET showed complete metabolic remission.

**Figure 5 diagnostics-13-01626-f005:**
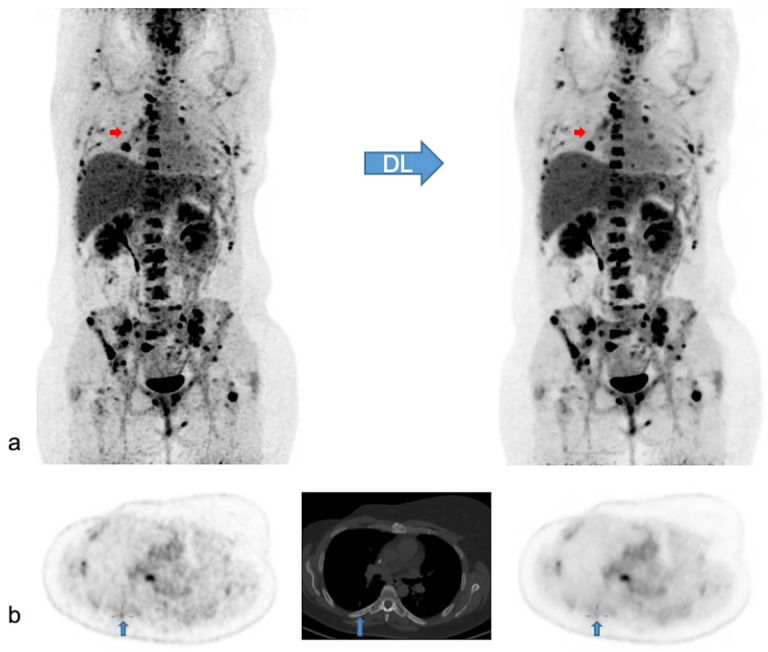
Native and denoised PET images of a patient with slightly better lesion detection with native PET. Legend: DL: deep learning processing by SubtlePET^TM^. A [^18^F]FDG PET/CT exam of a 37-year-old female patient, weight 105 kg, BMI 34 kg/m^2^ and FM 48 kg. She was referred for the follow-up of breast cancer with multiple (strongly) [^18^F]FDG avid bone and nodal metastases. In (**a**) MIP views are illustrated of native (on the left) and denoised PET (on the right). Almost all lesions were detected similarly in both PET modalities. Nonetheless, denoised images appeared less noisy, especially in the liver. Small red arrows in (**a**) depict a false negative small bone metastasis in denoised PET, further illustrated on axial PET/CT slices in (**b**) (upward blue arrows). This small low-activity focus corresponds to a lytic bone lesion measuring 4 mm on CT (**b**, upward blue arrows). This false negative lesion on denoised PET had no clinical impact.

**Figure 6 diagnostics-13-01626-f006:**
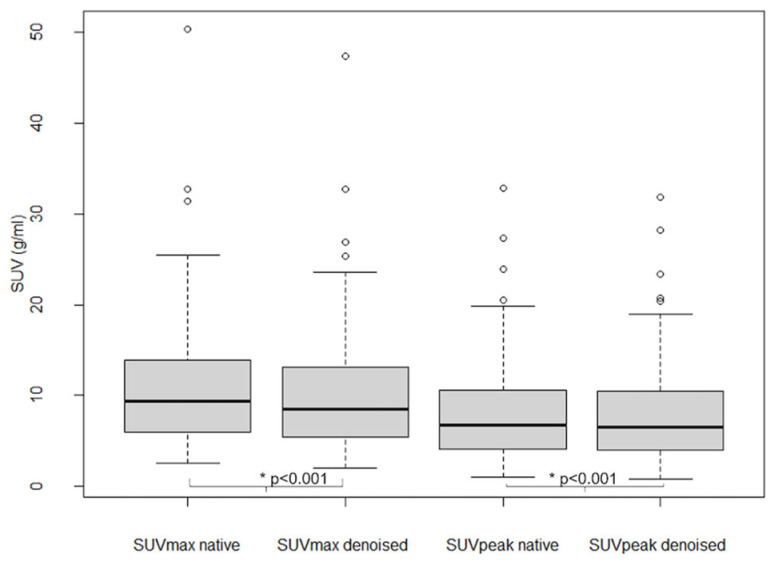
Lesion SUV_max_ and SUV_peak_ in native and denoised PET. Legend: Boxplots. Thick black lines in boxes show median values and upper and lower limits of boxes 25th and 75th percentiles. Also minimum and maximum values are indicated as horizontal lines, excluding outliers (shown as cercles).

## Data Availability

The datasets used and/or analysed during the current study are available from the corresponding author on reasonable request.

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
