# Peer review of "Deep Learning Denoising Improves and Homogenizes Patient [^18^F]FDG PET Image Quality in Digital PET/CT"

_diagnostics, 2023, doi:10.3390/diagnostics13091626_

Round 1
Reviewer 1 Report (Previous Reviewer 2)
I would advise the authors to lengthen the conclusion section in order to provide a clearer understanding of the comparative research and results between deep learning and assessments of the quality of PET/CT images.
Overall, the study is good enough to be accepted for publication.
Could be improved.
Author Response
Please see the attachment.

Reviewer 2 Report (Previous Reviewer 1)
Dear Authors,
I do appreciate your efforts in reviewing the manuscript. So for me now, it is suitable for the publication

Dear Editor,
the manuscript has been significantly revised, and I believe it is now ready for publication.
Author Response
Dear Authors,
I appreciate the time and effort you put into reviewing the manuscript.
Therefore, it is now suitable for publication in my opinion.
A: Thank you very much for your time and constructive comments which helped us to improve our work.
This manuscript is a resubmission of an earlier submission. The following is a list of the peer review reports and author responses from that submission.
Round 1
Reviewer 1 Report
Dear Authors,
minor revision are needed

Reviewer 2 Report
The paper titled ‘Deep learning denoising improves and homogenises patient [18F] FDG PET image quality in digital PET/CT’ tried to document uses of Subtle PETTM (a commercially available CNN-based software for PET) for improvement of IQ in patients and for IQ homogenization. This research paper lacks several essential elements that it should include. So, it appears to be more of an assignment. The work makes no methodological or original contributions to the research area. Overall, the paper is incomplete and cannot be published in its current state. Before it can be considered for publication, the paper needs to be drastically revised. The few significant problems with the paper are as follows.
1. Abstract is not complete; Authors presented only a goal and outcome in the abstract. Unfortunately, the following is absent from abstract: Your research's context or background; Your research question and what is known about it; justifications for the research, or why it is crucial to answer the question; strategies for answering the study topic and/or analytical strategies, justifications, and consequences of your findings.
2. What is ‘commercially available deep learning PET denoising solution’ in the abstract; is it AI software for PET?
3. Again, the introduction is incomplete and is missing a several key components of introduction section in the research paper.
4. Same in the methods section, it’s incomplete.
5. In this work, there is contribution to the scientific community is provided by authors.
6. Also, reviewer feels that many contents of the paper are not formatted or written like sentences.